# Peer video feedback builds basic life support skills: A randomized controlled non-inferiority trial

**Saša Sopka**[1,2]*, **Fabian Hahn**[1], **Lina Vogt**[1,2], **Kim Hannah Pears**[1], **Rolf Rossaint**[2], **Jenny Rudolph**[3], **Martin Klasen**[1]

**1** Medical Faculty, AIXTRA–Competency Center for Training and Patient Safety, RWTH Aachen University, Aachen, Germany, **2** Medical Faculty, Department of Anaesthesiology, University Hospital Aachen, RWTH Aachen University, Aachen, Germany, **3** Center for Medical Simulation, Boston, MA, United States of America

\* ssopka@ukaachen.de

## Abstract

### Introduction

Training Basic Life Support saves lives. However, current BLS training approaches are time-consuming and costly. Alternative cost-efficient and effective training methods are highly needed. The present study evaluated whether a video-feedback supported peer-guided Basic Life Support training approach achieves similar practical performance as a standard instructor-guided training in laypersons.

### Methods

In a randomized controlled non-inferiority trial, 288 first-year medical students were randomized to two study arms with different Basic Life Support training methods: 1) Standard Instructor Feedback (SIF) or 2) a Peer Video Feedback (PVF). Outcome parameters were objective data for Basic Life Support performance (compression depth and rate) from a resuscitation manikin with recording software as well as overall Basic Life Support performance and subjective confidence. Non-inferiority margins ($\Delta$) for these outcome parameters and sample size calculation were based on previous studies with Standard Instructor Feedback. Two-sided 95% confidence intervals were employed to determine significance of non-inferiority.

### Results

Results confirmed non-inferiority of Peer Video Feedback to Standard Instructor Feedback for compression depth (proportion difference PVF–SIF = 2.9%; 95% CI: -8.2% to 14.1%; $\Delta$ = -19%), overall Basic Life Support performance (proportion difference PVF–SIF = 6.7%; 95% CI: 0.0% to 14.3%; $\Delta$ = -27%) and subjective confidence for CPR performance (proportion difference PVF–SIF = -0.01; 95% CI: -0.18–0.17; $\Delta$ = -0.5) and emergency situations (proportion difference PVF–SIF = -0.02; 95% CI: -0.21–0.18; $\Delta$ = -0.5). Results for compression rate were inconclusive.

**Data Availability Statement:** All relevant data are within the manuscript and its Supporting Information files.

**Funding:** The authors received no specific funding for this work.

**Competing interests:** The authors have declared that no competing interests exist.

## Discussion

Peer Video Feedback achieves comparable results as standard instructor-based training methods. It is an easy-to-apply and cost-efficient alternative to standard Basic Life Support training methods. To improve performance with respect to compression rate, additional implementation of a metronome is recommended.

## Introduction

Teaching Basic Life Support (BLS) competencies saves lives. Optimally applied, BLS skills can save 30–40% of patients with sudden cardiac arrest until their admittance to acute hospitals for further treatment [1]. Since sudden cardiac arrest is one of the leading causes of death, [2,3] the demand for effective BLS training becomes even clearer. Thus, it has been recommended to teach both healthcare providers and laypersons in BLS [4]. Unfortunately, BLS skills are poor even among many health care professionals [5,6] and medical students [7]. As a consequence, it has been suggested that regular BLS trainings of at least 2 hours per year should be mandatory for all medical degrees [8]. Thus, medical schools and other qualifying institutions are increasingly interested in developing and accessing learners' BLS competencies. This focus is evidenced by substantial commitments to Entrustable Professional Activities (EPA) for medical professions in the US [9], and parallel emphasis on competency assessment across Europe and Canada [10–13].

BLS training approaches are built on guidelines of the European Resuscitation Council (ERC) and the American Heart Association (AHA). The latter also offer some recommended training approaches like the 4-step approach [14]. However, current BLS training approaches are time-consuming, costly, and require qualified staff, i.e. instructors [15].Thus, there is an urgent need for more resource-efficient and equally effective training methods.

The training of BLS competences is well possible with the methods established so far and already achieves satisfactory results in an elaborate small group setting. In order to achieve a large-scale qualification of laypersons as well as experts in this clinical competence, it is necessary to explore new didactic approaches. A recent European Resuscitation Council guidance note [16] suggests the use of peer-to-peer teaching as a cost-efficient alternative to current teaching formats. Implementing peer-guided learning and assessment in BLS training could enhance learners' skill acquisition while reducing expert instructor input. Peer-to-peer teaching has shown much promise, but it remains unclear how exactly it might play out in the context of learning BLS. In particular, it is unclear whether peer-guided learning is a robust substitute for expert-guided instruction. One option may be that peer teaching enables learners to better detect and correct gaps between BLS standard and their current performance through comparison, reflection, and self-initiated adjustment. Developing the facility to assess oneself and others using a rubric strengthens learners' conceptualization of the skill they are attempting to acquire. This happens in three ways: 1) developing a robust mental model of the core activity, 2) identifying common faults in the activity, and 3) ability to correctly identify key correct sub-behaviors of the activity [17]. Peer teaching covers these steps and is therefore a well-established concept for different clinical skills [6,18,19]. These and other studies found that learning outcomes of novice trainees were equal after BLS training led by peer tutors (i.e. student instructors trained in teaching BLS) and training led by staff members (physicians) as instructors [20,21].

In this study we investigated the suitability of a cost-efficient and easily applicable peer-to-peer teaching approach for BLS skills. Importantly, this approach should not be inferior to a

standard instructor-led training with respect to the essential learning outcomes. We thus compared a Peer Video Feedback (PVF) method, in which untrained co-learners provided feedback based on an ideal training video and self-created video recordings, with an established standard BLS training approach (4-step approach method with Standard Instructor Feedback, SIF) [14]. We hypothesized that training with peer video feedback was non-inferior to training with standard instructor feedback regarding BLS performance (compression depth and rate), adherence to the BLS algorithm, and self-reported confidence.

## Methods

### Ethics

Ethical approval for this study (Ethical Committee 2016/20) was provided by the Ethical Committee of the University Hospital, RWTH Aachen, Germany and designed according to the ethical principles of the World Medical Association [22].

### Participants

All participants were first-year students during the first three weeks of their medical studies at RWTH Aachen University, Germany. Data assessment took place during a mandatory introductory course on emergency medicine from 15[th] until 31[st] October 2014. To assure a homogeneous sample of BLS-naïve participants, previous educational background in healthcare was the only exclusion criterion. All participants signed an informed consent form.

### Study design

We conducted a randomized non-inferiority trial to compare two training methods for BLS skills in a parallel group design with two study arms. The goal was to determine whether a novel teaching method is non-inferior to an established one [23]The trial was designed to assess whether PVF was non-inferior to a conventional training approach (SIF).

Participants were randomly assigned to one of the study arms (PVF vs. SIF). To assure comparability of the training methods regarding complexity and training time, both followed a similar 4-step structure. Steps 1 to 3 were identical for both methods and taken from Peyton's 4-step approach [14]. Based on Albert Bandura's social learning theory [24], this approach is widely used for teaching BLS skills. Both training methods used a *Resusci Anne*[TM] manikin (Laerdal, Stavanger, Norway).

Step 1: Trained instructor demonstrated correct BLS performance without commenting.

Step 2: Additional demonstration with detailed step-by-step explanations.

Step 3: Instructor performed BLS guided by the learners.

For both methods, Steps 1 to 3 took place in large groups (60–80 persons). Both study arms differed with respect to the method applied in Step 4 during the following 90-minute training session:

**Arm 1: Standard Instructor Feedback (SIF) practical training.** Step 4 in this arm took place in groups of 10 persons. Since we could not find any recommendation in the literature regarding the group size, the latter was based on organizational aspects (number of medical students, availability of instructors, time frame, room availability etc.). All learners performed BLS on the manikin themselves and received individual feedback from a qualified instructor. Each participant performed at least two rounds of 2 minute lasting CPR. Furthermore, participants observed the other learners performing BLS and receiving feedback (corresponding to a

minimum of 36 minutes of observation). Step 4 in Arm 1 corresponded to Step 4 in Peyton's approach (7).

**Arm 2: Peer Video Feedback (PVF) practical training.** Step 4 took place in small teams of two persons with no instructor. First, both learners trained independently with the manikin, assisted by a checklist of quality characteristics. To give additional assistance for reaching the correct compression depth, the manikin gave an acoustic signal when the correct depth was achieved (which was not present in the SIF condition). The acoustic feedback was only present during training and not during skill assessment (please see below). A sample video of an ideal BLS performance was provided for reference and comparison to one's own performance. Subsequently, the partners filmed each other during BLS performance. They then analyzed the videos showing their own performance using the quality checklist. These activities were repeated until both partners had the subjective impression of having reached a good level of performance and confidence. Total individual training times were identical for both study arms. A study flow diagram based on the CONSORT guidelines can be seen in Fig 1. The study design is depicted in Fig 2.

## Skill assessment

Before (t0) and after (t1) the training, BLS performance was assessed using the same manikin for both study arms. Participants were instructed to imagine the manikin was a person

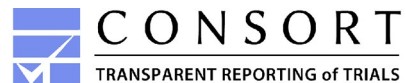

CONSORT 2010 Flow Diagram

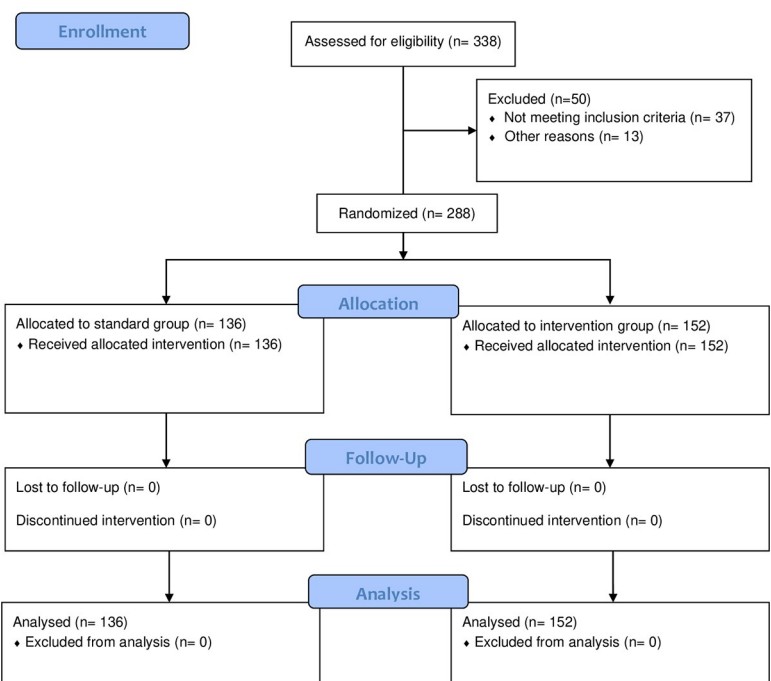

**Fig 1. CONSORT flow diagram.**

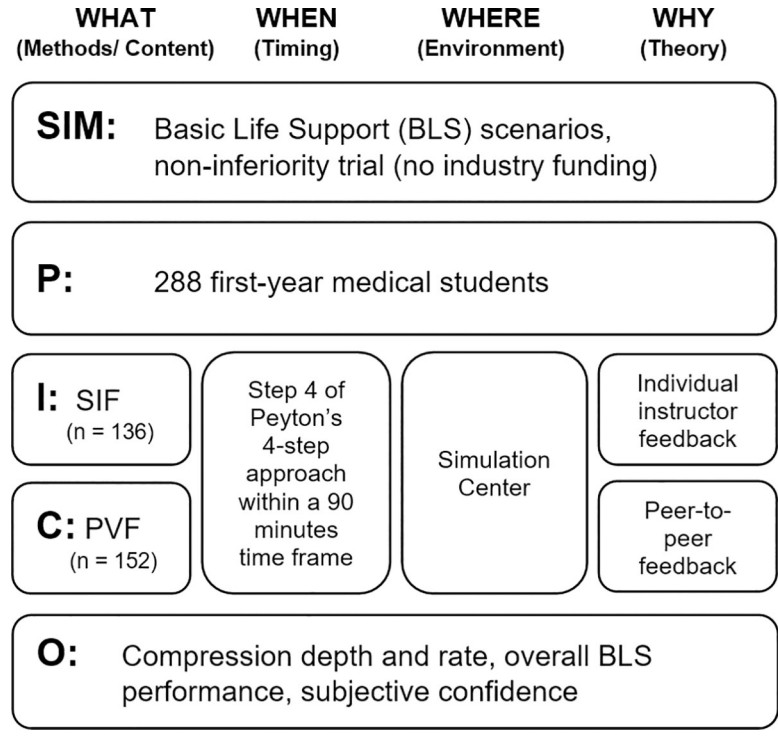

**Fig 2. SimPICO based on Raemer et al.** [25]. *Notes*. SIM: Simulation study, P: Population, I: Intervention, C: Comparator, O: Outcome; SIF: Standard Instructor Feedback; PVF: Peer Video Feedback.

collapsing next to them and to take all measures necessary. The scenario was terminated 120 seconds after the first chest compression. Compression depth and rate were recorded by the manikin's Laerdal PC SkillReporting Software.

Correct compression depth and rate after BLS training served as the main outcomes for the non-inferiority study. Based on the ERC guidelines [26], correct compression depth was defined as an average value between 50 and 59 mm. Correct compression rate was defined as an average rate of 100–120 compressions per minute.

Participants' performance regarding the BLS algorithm was assessed by an experienced rater via standardized checklist covering safe approach, control of consciousness and of breathing and emergency call. In line with previous research of our lab [27–29], correctness of the algorithm was coded if a participant performed more than 60% correctly.

## Confidence assessment

Participants rated their confidence a) during CPR and b) in an emergency situation with a non-responsive person. Ratings were obtained before and after training on a 6-point Likert scale (1 = "not at all confident", 6 = "very confident").

## Definition of non-inferiority margins

The definition of the non-inferiority margins was based on previous studies of our lab [27,28]. These studies investigated the rates of successful BLS performance (% of participants) after training with Peyton's 4-step approach for various samples of BLS-naïve subjects. Results from these studies covered a range of 19 percentage points for both compression depth (45% - 64%) and compression rate (33% - 52%). Since these are outcome variations within the standard

approach, any outcomes of another training method within these ranges can be considered as non-inferior. Thus, for the comparison of PVF and SIF, Δ = -19% was defined as non-inferiority margins for both compression depth and rate.

Non-inferiority margins for overall BLS performance were similarly based on the above-mentioned studies. Results yielded a range of 27 percentage points (65% - 92%). Therefore, Δ = -27% was defined as the non-inferiority margin for the BLS ratings. Since there was no comparable data available for confidence ratings, a difference of -0.5 points on the 6-point Likert scale was defined as the non-inferiority margin.

## Sample size planning

Sample size for non-inferiority testing was calculated after the method described by Blackwelder [30], using the Sealed Envelope Power Calculator [31]. Assuming an α significance level of .05 and a power (1-β) of 90%, the estimated sample size was N = 236 for compression depth and N = 234 for compression rate. To cover both main outcomes adequately, we decided for a minimum sample size of N = 236 (118 per group).

## Randomization

Before study begin, an independent administrative employee of the student's deanery, who was blinded to the study, randomized and allocated the students into groups of 10 persons. Allocation was performed following a sequence of random numbers and stratified by gender to realize a homogenous ratio. The groups were allocated into the study arms with a balanced scheme taking the facilities' room options into account.

## Statistical analysis

Data were analyzed with IBM SPSS Statistics Version 25 (IBM Corp., Armonk, NY, USA). Non-inferiority was assessed by comparing the percentage of successful performances (for compression depth and rate) after training in both study arms. According to the recommendations of the CONSORT statement [32], we employed two-sided 95% confidence intervals (CI) to determine significance of non-inferiority. Significance of results was given for 95% CIs of empirical percentage differences excluding the non-inferiority margin values [33,34]. CIs for the difference between percentages were calculated with the Wilson score interval method for independent proportions, which is free from aberrations and has good coverage properties [35]. Analogously, the 95% CI of the difference between the Likert scale confidence ratings in both study arms was used.

Raw data of the study underlying all analyses reported in this paper can be found under S1 Raw data.

# Results

## Sample characteristics

From 338 course participants, 37 were excluded from the analyses due to educational backgrounds in health care. Data from 13 additional participants were incomplete or missing, leaving a final sample of n = 288 (218 female, age 19.99 ± 3.75 years).

## Descriptive data

Pre- and post-training performance data and self-reported confidence ratings for the SIF group (N = 136; 104 female) and the PVF group (N = 152; 114 female) are reported in Table 1.

**Table 1. Performance data and self-reported confidence ratings before and after the training.**

| | Before training (t0) | | After training (t1) | |
|---|---|---|---|---|
| | SIF | PVF | SIF | PVF |
| Average compression depth in mm–median (IQR) | 39 (22) | 47 (19) | 55 (10) | 55.5 (10) |
| Average compression rate (compressions per minute)–median (IQR) | 101.5 (37) | 91 (40) | 110 (20) | 114 (24) |
| BLS algorithm (total IA points)–median (IQR) | 3 (2) | 2 (2) | 6 (2) | 7 (2) |
| Confidence for CPR performance–median (IQR) | 3 (2) | 3 (2) | 5 (0) | 5 (0) |
| Confidence for emergency situation–median (IQR) | 3 (2) | 2 (1) | 5 (1) | 5 (1) |

*Notes.* Medians and inter quartile ranges (IQR) before (t0) and after (t1) the training for average compression depth in mm, average compression rate in compressions per minute, BLS algorithm as measured by total IA points and self-reported confidence for CPR performance and in an emergency situation. t0: Pre-course test; t1: Post-course test; SIF: Standard Instructor Feedback; PVF: Peer Video Feedback.

## Non-inferiority tests

Results of non-inferiority analyses are depicted in Fig 3–5, which show the respective proportion differences between both training methods and 95% CI. Values <0 favor SIF and values >0 favor PVF. Blue lines indicate the respective non-inferiority margin (Δ).

**Compression depth.** After training, 62.5% of participants in the PVF group and 59.6% of participants in the SIF group achieved a correct compression depth, resulting in a proportion difference of 2.9% in favor of PVF (95% CI: -8.2% to 14.1%). Since the lower bound of the CI was above the inferiority margin of Δ = -19%, the results indicate a non-inferiority of PVF. The results are depicted in Fig 3.

**Compression rate.** After training, 42.1% of participants in the PVF group and 52.9% of participants in the SIF group achieved a correct compression rate, resulting in a proportion difference of -10.8% in favor of SIF (95% CI: -22.0% to 0.1%). Since the lower bound of the CI was below the non-inferiority margin of Δ = -19%, a non-inferiority of PVF could not be confirmed for compression rate. The results of this comparison thus remain inconclusive (Fig 3).

**BLS performance.** After training, 92.8% of participants in the PVF group and 86.0% of participants in the SIF group achieved a correct BLS algorithm performance, resulting in a proportion difference of 6.7% in favor of PVF (95% CI: 0.0% to 14.3%). Since the lower bound of the CI was above the non-inferiority margin of Δ = -27%, the results indicate a non-inferiority of PVF. The results are depicted in Fig 4.

**Confidence ratings.** Mean differences between the two groups were -0.01 (95% CI: -0.18–0.17) for confidence for CPR performance and -0.02 (95% CI: -0.21–0.18) for confidence for

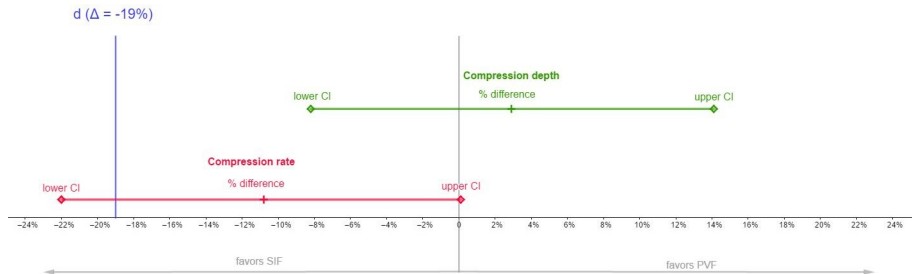

**Fig 3. Non-inferiority results for compression depth and rate.** *Notes.* Proportion (%) difference, lower and upper 95% CI for both compression rate and compression depth. The line marked "d" indicates the non-inferiority margin. Values <0 favor SIF and values >0 favor PVF. Significant non-inferiority is given when the lower CI lies above d.

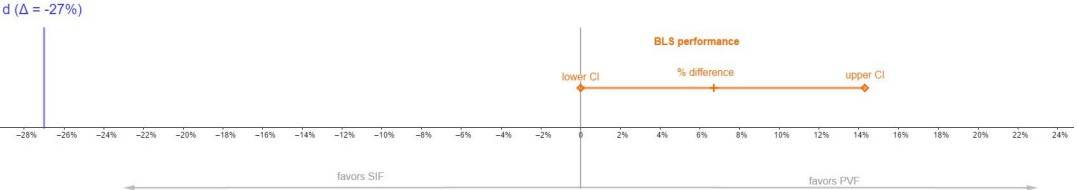

**Fig 4. Non-inferiority results for overall BLS performance as measured with the BLS algorithm checklist.** *Notes.*
Proportion (%) difference, lower and upper 95% CI for BLS performance. The line marked "d" indicates the non-inferiority
margin. Values <0 favor SIF and values >0 favor PVF. Significant non-inferiority is given when the lower CI lies above d. BLS:
Basic life support.

emergency situations with a non-responding person. Since the lower bound of the CI was
above the non-inferiority margin of Δ = -0.5, the results indicate a non-inferiority of PVF for
both items. The results are depicted in Fig 5.

## Discussion

In the present study, we investigated the non-inferiority of a BLS training method based on
video-assisted peer-to-peer feedback as compared to an established standard. The results con-
firmed non-inferiority of the peer-guided method regarding compression depth, performance
of the BLS algorithm, and subjective confidence. Findings on compression rate remained
inconclusive. In summary, the PVF method achieved comparable performance in several
domains and may constitute a promising alternative to instructor-based small group concepts.

A clear strength of the PVF approach is its low cost and low infrastructural requirements.
Besides a room with a manikin, the training concept only requires written instructions, an
instruction video, and a tablet or smartphone with a camera. Even in larger groups of learners,
no qualified instructor is needed. Given the wide distribution of smartphones, these demands
can be easily met. Participants can be laypersons without prior experience in BLS. The great
advantage of this training is the saving of qualified experienced instructors, who are often a
rare resource when it comes to trainings in schools, sports clubs, or companies. This sugges-
tion is in line with a recent ERC guidance note [36] highlighting a potential role of peer-teach-
ing healthcare students as multipliers for BLS skills. This aspect is particularly important since
a broad BLS education has been recommended for the general population from early school

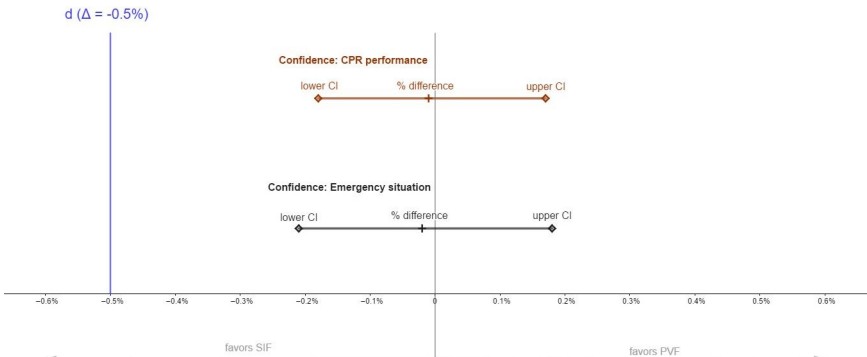

**Fig 5. Non-inferiority results for self-reported confidence.** *Notes.* Proportion (%) difference, lower and upper 95%
CI for both confidence regarding CPR performance and confidence regarding an emergency situation. The line
marked "d" indicates the non-inferiority margin. Values <0 favor SIF and values >0 favor PVF. Significant non-
inferiority is given when the lower CI lies above d. CPR: Cardiopulmonary resuscitation.

age on [37]. In Europe, 16 countries officially suggest CPR education in schools, and five even have a legislation on this issue [38]. In Germany, a four-hour teaching unit is recommended annually for children 12 years and older [39]. However, the degree of implementation still varies from region to region, which highlights the necessity to make BLS trainings more accessible. PVF makes it easier to offer BLS trainings in such contexts and could convey BLS skills to large parts of the population. PVF is highly standardized and can minimize variations in the instruction quality. Indeed, past research has shown that instructor-based training in small groups is often unstandardized, varies regarding instructions and feedback, and can result in poor skill retention [40]. Despite the fact that the approach is promising, it must be noted that the video feedback method may not be suitable for every learner. In particular, this method may primarily address persons who are already familiar with digital media, e.g. the use of smartphones.

While our study focuses on BLS outcomes, it seems plausible that the collaborative work of learning and discussing may produce secondary benefits of peer feedback learning. In specific, these may be relationship and trust building that could enhance medical school experience for our learners. Rather than seeing each other as competitors for scarce instructor time, these experiences may build students' sense of a shared purpose by learning in a collaborative approach, fostering seeking feedback and giving skills.

Previous studies demonstrate that the willingness of laypersons to perform bystander BLS improves through undergoing BLS training [41,42]. An important psychological factor contributing to this effect may be the level of subjective confidence for performing CPR [43]. In our study, PVF was non-inferior to standard training regarding both investigated confidence dimensions. Thus, it is reasonable to assume that PVF is equally suited to increase the willingness and readiness of laypersons.

Although our study found no significant difference between the training methods with respect to compression rate, the inconclusive results for this domain highlight a potential for improvement of PVF. Maintaining a predefined rate of compressions may be facilitated by a metronome. Since there are various online metronome applications for smartphones or tablets available, this could be easily implemented in PVF. Another promising approach may be the combination of auditory cues such as metronome beats with real-time visual feedback (such as in [44]). Multimodal sensory input may well improve training results further and should be the subject of future research.

Even though the reduction in instructors will likely allow for significantly more BLS training to be offered area-wide, the BLS training manikins may be a limiting factor. A way out of this dilemma may be producing and providing an increased number of very simple and inexpensive simulators. Another approach may be temporal rotation systems for manikins. Using digital booking systems, the latter would be relatively easy to achieve, especially since well-developed technical solutions already exist in other fields (e.g. car sharing).

## Limitations

The present study has some limitations that need to be acknowledged. First of all, more findings from other contexts are required to confirm our results. Specifically, further investigations are needed to show whether these findings can be generalized to other populations of medical laypersons. Our sample for this study consisted of medical students. This group is not representative for the general population in several ways. First, they were medically interested; thus, it is reasonable to assume that they were more motivated than a general audience to learn first aid skills. Second, they were most likely of high intelligence and equipped with more mental strategies (e.g., scientific thinking and culture) than the average population, which probably

fostered understanding videos and partner's suggestions and facilitated medical skills learning. Finally, they were very young, thus they could easily perform CPR due to physical fitness. Further research is needed to confirm whether PVF can be equally well applied in older participants with other educational background.

Another limitation for the transferability of the results could be the chosen group size in the instructor-led group. It can be strongly assumed that in a BLS training the ratio of instructor to participant has a noticeable influence on the learning success. For this reason, our findings still have to be confirmed in other settings where a smaller instructor-participant ratio is offered. In general, however, it can be stated that participants in our study consistently show satisfactory results in both studies with regard to the target values recommended by the international guidelines.

Moreover, the teaching methods required different group sizes (2 vs. 10) in the PVF and SIF study arms. It must be considered that small groups may be necessary when applying feedback of untrained peers in order to ensure adequate time for and quality of feedback, which is why we deliberately chose to keep the groups small in the PVF study arm. However, we cannot exclude the possibility that the different group sizes may have introduced a bias. Nevertheless, the effect could only be studied under training laboratory conditions, since in clinical reality, where resources are limited, it is difficult to imagine teaching 2 people with one instructor.

In summary, the PVF approach is an easy-to-apply and cost-efficient alternative to standard BLS training methods. To improve performance with respect to compression rate, the implementation of a metronome is recommended. Further development of this method could significantly contribute to improve the populations' BLS skills.

## Supporting information

**S1 Checklist. CONSORT statement 2006—checklist for non-inferiority and equivalence trials.**
(DOC)

**S1 Raw data. Raw data underlying the analyses.** The data set contains the following:

- basic demographic data of the participants (age, sex)

- study group assignment of the participants (1 = standard instructor feedback, 2 = peer video feedback)

- raw data for time points t0 and t1:

- Confidence ratings (2 items: 1. feeling safe to do chest compression and 2. feeling safe in an emergency situation with a non-responding person)

- average compression depth (2 variables: 1. subject-wise average values and 2. categorial coding if the guidelines were met or not)

- average compression rate (2 variables: 1. subject-wise average rate (compressions per minute) and 2. categorial coding if the guidelines were met or not)

- performance on the BLS algorithm ((2 variables: 1. subject-wise points and 2. categorial coding if 60% of attainable points were achieved or not).
(XLSX)

## Author Contributions

**Conceptualization:** Saša Sopka, Fabian Hahn.

**Data curation:** Saša Sopka, Fabian Hahn, Kim Hannah Pears.

**Formal analysis:** Saša Sopka, Fabian Hahn, Martin Klasen.

**Investigation:** Saša Sopka, Lina Vogt, Kim Hannah Pears.

**Methodology:** Saša Sopka, Fabian Hahn, Kim Hannah Pears, Martin Klasen.

**Project administration:** Saša Sopka.

**Resources:** Saša Sopka.

**Supervision:** Saša Sopka, Rolf Rossaint, Jenny Rudolph, Martin Klasen.

**Writing – original draft:** Saša Sopka, Fabian Hahn.

**Writing – review & editing:** Saša Sopka, Lina Vogt, Kim Hannah Pears, Rolf Rossaint, Jenny Rudolph, Martin Klasen.

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
