## [Decision Letter · Decision Letter 0]

1 Apr 2021

PONE-D-21-07914

Peer video feedback builds basic life support skills: 

A randomized controlled non-inferiority trial reveals live-saving performance with less instructor input

PLOS ONE

Dear Dr.Sasa Sopka

Thank you for submitting your manuscript to PLOS ONE. After careful consideration, we feel that it has merit but does not fully meet PLOS ONE’s publication criteria as it currently stands. Therefore, we invite you to submit a revised version of the manuscript that addresses the points raised during the review process.

Thank you very much for having submitted this paper. It's interesting and easy to go through. However it shows some limitation and weaknesses as emerged from the comments of the reviewers who are experts in such a filed.

I think that that following their indications your paper could improve in clarity and quality.

We look forward to receiving your revised manuscript.

Kind regards,

Simone Savastano

Academic Editor

PLOS ONE

Additional Editor Comments:

Thank you very much for having submitted this paper. It's interesting and easy to go through. However it shows some limitation and weaknesses as emerged from the comments of the reviewers who are experts in such a filed.

I think that that following their indications your paper could improve in clarity and quality.

Journal Requirements:

Please consider changing the title so as to meet our title format requirement (https://journals.plos.org/plosone/s/submission-guidelines). In particular, the title should be "Specific, descriptive, concise, and comprehensible to readers outside the field" and in this case it is not informative and specific about your study's scope; namely, the students did not saved lives properly speaking for this study.

We note that you have indicated that data from this study are available upon request. PLOS only allows data to be available upon request if there are legal or ethical restrictions on sharing data publicly. For information on unacceptable data access restrictions, please see http://journals.plos.org/plosone/s/data-availability#loc-unacceptable-data-access-restrictions.

3a) If there are ethical or legal restrictions on sharing a de-identified data set, please explain them in detail (e.g., data contain potentially identifying or sensitive patient information) and who has imposed them (e.g., an ethics committee). Please also provide contact information for a data access committee, ethics committee, or other institutional body to which data requests may be sent.

3b) If there are no restrictions, please upload the minimal anonymized data set necessary to replicate your study findings as either Supporting Information files or to a stable, public repository and provide us with the relevant URLs, DOIs, or accession numbers. Please see http://www.bmj.com/content/340/bmj.c181.long for guidelines on how to de-identify and prepare clinical data for publication. For a list of acceptable repositories, please see http://journals.plos.org/plosone/s/data-availability#loc-recommended-repositories.

Please include captions for your Supporting Information files at the end of your manuscript, and update any in-text citations to match accordingly. Please see our Supporting Information guidelines for more information: http://journals.plos.org/plosone/s/supporting-information.

Reviewers' comments:

Reviewer's Responses to Questions

**Comments to the Author**

1. Is the manuscript technically sound, and do the data support the conclusions?

Reviewer #1: Yes

Reviewer #2: Yes

2. Has the statistical analysis been performed appropriately and rigorously? 

Reviewer #1: Yes

Reviewer #2: Yes

3. Have the authors made all data underlying the findings in their manuscript fully available?

Reviewer #1: Yes

Reviewer #2: Yes

4. Is the manuscript presented in an intelligible fashion and written in standard English?

Reviewer #1: Yes

Reviewer #2: Yes

5. Review Comments to the Author

Reviewer #1: Dear authors,

thank you for submitting your manuscript. I believe it is well written and organized, and face an important aspect of CPR training. However, I have some concerns that I hope you can address improving the quality of your manuscript.

Abstract

- I would prefer to have an abstract divided in the classical sub-headings for improve clarity

- I suggest to add some numeric results to the abstract

Introduction

- I believe you missed to consider the most recent and wide survey regarding medical students' knowledge about CPR and cardiac arrest (Int J Cardiol . 2019 Dec 1;296:76-80. doi: 10.1016/j.ijcard.2019.07.016)

- I suggest to have a look to the recent ERC guidance note regarding healthcare students training in CPR (Eur J Anaesthesiol

. 2020 Oct;37(10):839-841. doi: 10.1097/EJA.0000000000001272.). In that document it is also specified that peer-to-peer could be a good strategy to teaching healthcare students, therefore it could be useful to introduce your idea

- I believe that the strategy to teach people in BLS basically comes from the ILCOR COSTR documents (Resuscitation. 2020 Nov;156:A188-A239. doi: 10.1016/j.resuscitation.2020.09.014. and Circulation. 2020 Oct 20;142(16_suppl_1):S222-S283. doi: 10.1161/CIR.0000000000000896.)

- You state that there is not a gold standard for BLS training. I believe it is not all thruth. I believe that the reasons for the usefulness of your experimentations (and the others regarding this topic) reside in the need to improve BLS teaching quality, involving more and more people without increasing cost, as you highlighted in another part of the discussion, rather than the lack of a gold standard for teaching

- line 87-88: I believe that this part pertains to methods section

Methods

- The standard classroom for training is usually 6 attendees for 1 instructors maximum. Why did you chose 10 participants for each instructors?

- line 132: who created the video? Is it following ERC or AHA guidelines?

Results

- Table 1: did you presented means and SD because all the variables had normal distribution? If not it could be better to present as median [IQR]

Discussion:

- line 327: compressione rate is better than frequency

- I suggest that another point of dicussion is to add the visual real-time feedback to PVF. There are pleny of demonstration regarding the fact that real-time visual feedback during training improves CPR performance, therefore a possibile future step could be to mix both. This can also help to improve the compression rate

- I believe that another limitation is that you decided to include more than 6 participants for each standard group and this could have affected the CPR quality in that group

Reviewer #2: The author investigates the non-inferiority of a new method to teach BLS that allows saving resources like instructors; this is a fundamental research topic to increase the number of trained people and save lives.

The paper lacks a "limitation" section, notwithstanding some limitation of the study, first of a selection bias: despite the "BLS-naive" feature, participants are from a medical class, so they are certainly more motivated than a general audience to learn first aid skills. Moreover, medical students may have more instruments (i.e., scientific thinking and culture) to understand videos and the partner's suggestions. In addition, the sample doesn't reflect a general audience about the age, where younger students can easily perform an acceptable CPR thanks to their fitness. Still, older people may take-advantage of expert teacher suggestions about tips-and-trick to execute chest compressions.

In line 151, the author declares that skill assessment was performed with the same manikin for both groups but that an acoustic feedback confirms the reach of a correct depth for every chest compression. In my opinion, this is a fundamental methodological error that could conceal differences between different teaching strategies: skill assessment should reflect reality, and groups could react differently in a situation where there is no external help.

In conclusion, this paper investigates an interesting topic, but I suggest to detail conclusions better accounting limitations.

6. PLOS authors have the option to publish the peer review history of their article (what does this mean?). If published, this will include your full peer review and any attached files.

Reviewer #1: No

Reviewer #2: No

---

## [Author Response · Author response to Decision Letter 0]

19 May 2021

The unformatted respond text is listed below. The original formatted and correct respond to the Editor/Reviewers is uploaded in the files section as a Word file. 

****

We thank the editors and the reviewers for their positive and constructive evaluation of our work. We modified the manuscript according to their comments and we are confident that they helped us to improve the paper substantially. Please find a pointwise reply to the comments below. All changes in the revised manuscript are highlighted in color. 

Journal Requirements:

Reply: We checked the manuscript again and made all necessary changes to meet PLOS ONE’s style requirements. Specifically, the file naming was changed. 

2. Please consider changing the title so as to meet our title format requirement (https://journals.plos.org/plosone/s/submission-guidelines). In particular, the title should be "Specific, descriptive, concise, and comprehensible to readers outside the field" and in this case it is not informative and specific about your study's scope; namely, the students did not saved lives properly speaking for this study.

Reply: We changed the title according to the editor’s suggestions and according to the title format requirement. The new title is as follows:

“Peer video feedback builds basic life support skills: A randomized controlled non-inferiority trial”

3b) If there are no restrictions, please upload the minimal anonymized data set necessary to replicate your study findings as either Supporting Information files or to a stable, public repository and provide us with the relevant URLs, DOIs, or accession numbers. Please see http://www.bmj.com/content/340/bmj.c181.long for guidelines on how to de-identify and prepare clinical data for publication. For a list of acceptable repositories, please see http://journals.plos.org/plosone/s/data-availability#loc-recommended-repositories.

Reply: We have included the minimal anonymized data set as Supporting Information in the revised version of the manuscript. 

Reply: A reference to the Supporting Information has been inserted at the end of the Methods section. Captions for the Supporting Information have been inserted at the end of the manuscript file.

Reviewers' comments:

Reviewer #1: 

Dear authors, thank you for submitting your manuscript. I believe it is well written and organized, and face an important aspect of CPR training. However, I have some concerns that I hope you can address improving the quality of your manuscript.

Abstract

1. - I would prefer to have an abstract divided in the classical sub-headings for improve clarity

Reply: We agree and we re-structured the abstract according to the IMRAD scheme (Introduction, Methods, Results And Discussion).

2. I suggest to add some numeric results to the abstract

Reply: We agree that numeric results can be helpful for the reader already in the abstract. We added numerical results of the non-inferiority analyses. The new abstract reads as follows:

“Introduction 

Training Basic Life Support saves lives. However, current BLS training approaches are time-consuming and costly. Alternative cost-efficient and effective training methods are highly needed. The present study evaluated whether a video-feedback supported peer-guided Basic Life Support training approach achieves similar practical performance as a standard instructor-guided training in laypersons. 

Methods

In a randomized controlled non-inferiority trial, 288 first-year medical students were randomized to two study arms with different Basic Life Support training methods: 1) Standard Instructor Feedback (SIF) or 2) a Peer Video Feedback (PVF). Outcome parameters were objective data for Basic Life Support performance (compression depth and rate) from a resuscitation manikin with recording software as well as overall Basic Life Support performance and subjective confidence. Non-inferiority margins (Δ) for these outcome parameters and sample size calculation were based on previous studies with Standard Instructor Feedback. Two-sided 95% confidence intervals were employed to determine significance of non-inferiority. 

Results

Results confirmed non-inferiority of Peer Video Feedback to Standard Instructor Feedback for compression depth (proportion difference PVF – SIF = 2.9%; 95% CI: -8.2% to 14.1%; Δ = -19%), overall Basic Life Support performance (proportion difference PVF – SIF = 6.7%; 95% CI: 0.0% to 14.3%; Δ = -27%) and subjective confidence for CPR performance (proportion difference PVF – SIF = -0.01; 95% CI: -0.18 – 0.17; Δ = -0.5) and emergency situations (proportion difference PVF – SIF = -0.02; 95% CI: -0.21 – 0.18; Δ = -0.5). Results for compression rate were inconclusive. 

Discussion

Peer Video Feedback achieves comparable results as standard instructor-based training methods. It is an easy-to-apply and cost-efficient alternative to standard Basic Life Support training methods. To improve performance with respect to compression rate, additional implementation of a metronome is recommended.”

Introduction

- I believe you missed to consider the most recent and wide survey regarding medical students' knowledge about CPR and cardiac arrest (Int J Cardiol . 2019 Dec 1;296:76-80. doi: 10.1016/j.ijcard.2019.07.016) 

Reply: Thank you for notifying us of this important work. We agree that this source should be included in the manuscript. Therefore, we included the following sentence in the Introduction: 

“Unfortunately, BLS skills are poor even among many health care professionals [4,5] and medical students [6]. As a consequence, it has been suggested that regular BLS trainings of at least 2 hours per year should be mandatory for all medical degrees (Baldi et al., 2019).” [...] 

- I suggest to have a look to the recent ERC guidance note regarding healthcare students training in CPR (Eur J Anaesthesiol

. 2020 Oct;37(10):839-841. doi: 10.1097/EJA.0000000000001272.). In that document it is also specified that peer-to-peer could be a good strategy to teaching healthcare students, therefore it could be useful to introduce your idea 

Reply: This is an excellent idea. We added the following sentence to our introduction:

“A recent European Resuscitation Council guidance note (Baldi et al., 2020) suggests the use of peer-to-peer teaching as a cost-efficient alternative to current teaching formats. Implementing peer-guided learning and assessment in BLS training could enhance learners’ skill acquisition while reducing expert instructor input.” [...]

Moreover, we found the idea of peer-teaching healthcare students as multipliers for BLS skills, as outlined in the guidance note, intriguing and included it in our Discussion section as follows:

“The great advantage of this training is the saving of qualified experienced instructors, who are often a rare resource when it comes to trainings in schools, sports clubs, or companies. This suggestion is in line with a recent ERC guidance note (Baldi et al., 2020) highlighting a potential role of peer-teaching healthcare students as multipliers for BLS skills.”

- I believe that the strategy to teach people in BLS basically comes from the ILCOR COSTR documents (Resuscitation. 2020 Nov;156:A188-A239. doi: 10.1016/j.resuscitation.2020.09.014. and Circulation. 2020 Oct 20;142(16_suppl_1):S222-S283. doi: 10.1161/CIR.0000000000000896.) 

Reply: We thank the reviewer for the hint and have included this important source in our introduction as follows:

“Since sudden cardiac arrest is one of the leading causes of death, [2,3] the demand for effective BLS training becomes even clearer. Thus, it has been recommended to teach both healthcare providers and laypersons in BLS (Greif et al., 2020). 

- You state that there is not a gold standard for BLS training. I believe it is not all thruth. I believe that the reasons for the usefulness of your experimentations (and the others regarding this topic) reside in the need to improve BLS teaching quality, involving more and more people without increasing cost, as you highlighted in another part of the discussion, rather than the lack of a gold standard for teaching 

Reply: We think that the reviewer makes a valid point here. Also, we think that the reviewer is entirely correct when highlighting that the motivation for our approach is easier access for more persons at lower costs. Therefore, we changed the corresponding passage as follows:

The training of BLS competences is well possible with the methods established so far and already achieves satisfactory results in an elaborate small group setting. In order to achieve a large-scale qualification of laypersons as well as experts in this clinical competence, it is necessary to explore new didactic approaches.

- line 87-88: I believe that this part pertains to methods section 

We totally agree and have moved the description to the methods section “study design” lines 127/128 

Methods

- The standard classroom for training is usually 6 attendees for 1 instructors maximum. Why did you choose 10 participants for each instructors? à 

Reply: Thank you very much for this important point. We will be happy to explain the motivation behind our approach. 

For many years, we have been conducting BLS training for different learner groups at our medical faculty. The students are assigned to different large and small groups with regard to clinical practice, such as our BLS training, which make clinical teaching at all possible. As we did not find any evidence-based recommendations for Instructor-led BLS training group size even after multiple searches, we decided for the most feasible group size (10) in the light of organizational aspects (number of medical students, availability of instructors, time frame, room availability etc.). This group size is most likely to be transferable to other curricula based on our experience over many years. Furthermore, even in the literature cited in the International Resuscitation Guidelines, we could not find a clearly defined group size for instructor-led BLS training (group sizes between 6-15).

We have therefore chosen this group size for practical reasons and over 15 years of experience with BLS training concepts in the „small" group with 10 participants. The clearly defined outcome parameters such as pressure depth, pressure frequency, and adherence to the BLS algorithm confirm our training results, which meet the international guidelines.

To explain our motivation also in the manuscript, we added the following text passage to the description of the SIF practical training:

“Since we could not find any recommendation in the literature regarding the group size, the latter was based on organizational aspects (number of medical students, availability of instructors, time frame, room availability etc.).”

- line 132: who created the video? Is it following ERC or AHA guidelines? 

Reply: Our team created the video with the help of experts for Audio-Video productions. It was created on the following the recommendations of the international ERC Resuscitation Guidelines.

Results

- Table 1: did you presented means and SD because all the variables had normal distribution? If not it could be better to present as median [IQR] 

Reply: Thank you for notifying us of this important aspect. Indeed, our data did not follow a normal distribution. Thus, we replaced the values in Table 1 by medians and interquartile ranges, as suggested by the reviewer. The new Table 1 is as follows:

 Before training (t0) After training (t1)

 SIF PVF SIF PVF

Average compression depth in mm – median (IQR) 39 (22) 47 (19) 55 (10) 55.5 (10)

Average compression rate (compressions per minute) – median (IQR) 101.5 (37) 91 (40) 110 (20) 114 (24)

BLS algorithm (total IA points) – median (IQR) 3 (2) 2 (2) 6 (2) 7 (2)

Confidence for CPR performance – median (IQR) 3 (2) 3 (2) 5 (0) 5 (0)

Confidence for emergency situation – median (IQR) 3 (2) 2 (1) 5 (1) 5 (1)

Notes. Medians and inter quartile ranges (IQR) before (t0) and after (t1) the training for average compression depth in mm, average compression rate in compressions per minute, BLS algorithm as measured by total IA points and self-reported confidence for CPR performance and in an emergency situation. t0: pre-course test; t1: post-course test; SIF: Standard Instructor Feedback; PVF: Peer Video Feedback.

Discussion:

- line 327: compressione rate is better than frequency 

Reply: This was indeed an inconsistent terminology. We have changed the wording to “compression rate”. 

- I suggest that another point of dicussion is to add the visual real-time feedback to PVF. There are pleny of demonstration regarding the fact that real-time visual feedback during training improves CPR performance, therefore a possibile future step could be to mix both. This can also help to improve the compression rate à Sasa

Reply: Thank you for this important suggestion. We have added this point to our discussion as follows: 

 Another promising approach may be the combination of auditory cues such as metronome beats with real-time visual feedback (such as in Baldi et al., 2017). Multimodal sensory input may well improve training results further and should be the subject of future research.

- I believe that another limitation is that you decided to include more than 6 participants for each standard group and this could have affected the CPR quality in that group 

Reply: This is a valid point, and we agree with the reviewer that the group size may have an influence on the study results. To acknowledge this fact in the revised manuscript, the following passage has been added to the Limitations section: 

“Another limitation for the transferability of the results could be the chosen group size in the instructor-led group. It can be strongly assumed that in a BLS training the ratio of instructor to participant has a noticeable influence on the learning success. For this reason, our findings still have to be confirmed in other settings where a smaller instructor-participant ratio is offered. In general, however, it can be stated that participants in our study consistently show satisfactory results in both studies with regard to the target values recommended by the international guidelines.”

Reviewer #2: 

The author investigates the non-inferiority of a new method to teach BLS that allows saving resources like instructors; this is a fundamental research topic to increase the number of trained people and save lives.

The paper lacks a "limitation" section, notwithstanding some limitation of the study, first of a selection bias: despite the "BLS-naive" feature, participants are from a medical class, so they are certainly more motivated than a general audience to learn first aid skills. Moreover, medical students may have more instruments (i.e., scientific thinking and culture) to understand videos and the partner's suggestions. In addition, the sample doesn't reflect a general audience about the age, where younger students can easily perform an acceptable CPR thanks to their fitness. Still, older people may take-advantage of expert teacher suggestions about tips-and-trick to execute chest compressions.

Reply: This is a valid point. To address the limitations raised by the reviewer, we introduced a new Limitations section. The respective passage about the sample reads as follows:

“Our sample for this study consisted of medical students. This group is not representative for the general population in several ways. First, they were medically interested; thus, it is reasonable to assume that they were more motivated than a general audience to learn first aid skills. Second, they were most likely of high intelligence and equipped with more mental strategies (e.g., scientific thinking and culture) than the average population, which probably fostered understanding videos and partner’s suggestions and facilitated medical skills learning. Finally, they were very young, thus they could easily perform CPR due to physical fitness. Further research is needed to confirm whether PVF can be equally well applied in older participants with other educational background.”

In line 151, the author declares that skill assessment was performed with the same manikin for both groups but that an acoustic feedback confirms the reach of a correct depth for every chest compression. In my opinion, this is a fundamental methodological error that could conceal differences between different teaching strategies: skill assessment should reflect reality, and groups could react differently in a situation where there is no external help. 

Reply: 

Thank you very much for this important feedback. It is absolutely correct that an assessment with an acoustic feedback would mean a methodological error, and this was neither planned by us nor was it carried out in this way in the study. We recognized that the text passage about the acoustic feedback was erroneously misplaced in the Skills Assessment section; instead, it was supposed to be located in the previous passage describing the PVF study arm. Our sincere apologies that this error was overlooked by all authors despite multiple readings of the text. Please find a detailed (and correct) explanation of the procedure below. The skills assessment was conducted with the same manikin for both arms of the study. This assessment manikin was similar but not exactly the same as in the training sessions; the latter was again identical for both study arms. The differences between the training and assessment manikins were negligible because all manikins were from the same manufacturer and the same product line. For the training sessions in the PVF group, the manikin provided an audible signal when the correct compression depth was reached. This additional real-time audio feedback was an element of the PVF training concept. The acoustic feedback was neither present in the training sessions of the SIF group nor in any assessment session of both study arms. 

We have removed the corresponding text passage from the Skill Assessment section and added the following description to the description of Study Arm 2 (PVF): 

“To give additional assistance for reaching the correct compression depth, the manikin gave an acoustic signal when the correct depth was achieved (which was not present in the SIF condition). The acoustic feedback was only present during training and not during skill assessment (please see below).” [...]

In conclusion, this paper investigates an interesting topic, but I suggest to detail conclusions better accounting limitations.

Reply: We thank the reviewer for the positive appraisal of our study topic, and we hope that we could sufficiently address the study’s limitations in the revised manuscript.

---

## [Decision Letter · Decision Letter 1]

7 Jul 2021

Peer video feedback builds basic life support skills: 

A randomized controlled non-inferiority trial

PONE-D-21-07914R1

Dear Dr. Sopka,

We’re pleased to inform you that your manuscript has been judged scientifically suitable for publication and will be formally accepted for publication once it meets all outstanding technical requirements.

Kind regards,

Simone Savastano

Academic Editor

PLOS ONE

Additional Editor Comments (optional):

Thank you very much for having improved the quality of the manuscript following the advice of the reviewers. You have successfully addressed all their concerns.

Reviewers' comments:

Reviewer's Responses to Questions

**Comments to the Author**

1. If the authors have adequately addressed your comments raised in a previous round of review and you feel that this manuscript is now acceptable for publication, you may indicate that here to bypass the “Comments to the Author” section, enter your conflict of interest statement in the “Confidential to Editor” section, and submit your "Accept" recommendation.

Reviewer #1: All comments have been addressed

Reviewer #2: (No Response)

2. Is the manuscript technically sound, and do the data support the conclusions?

Reviewer #1: Yes

Reviewer #2: (No Response)

3. Has the statistical analysis been performed appropriately and rigorously? 

Reviewer #1: Yes

Reviewer #2: (No Response)

4. Have the authors made all data underlying the findings in their manuscript fully available?

Reviewer #1: Yes

Reviewer #2: (No Response)

5. Is the manuscript presented in an intelligible fashion and written in standard English?

Reviewer #1: Yes

Reviewer #2: (No Response)

6. Review Comments to the Author

Reviewer #1: Dear authors, thank you for addressing my comments. I have no further concerns regarding your manuscript.

Reviewer #2: Thanks for the reply, most of my doubt were solved, just some suggestion:

- line 100: there is a double dot at the end of the sentence

- line 150: now it's clear, I suggest to remove the "(please see below)"

7. PLOS authors have the option to publish the peer review history of their article (what does this mean?). If published, this will include your full peer review and any attached files.

Reviewer #1: No

Reviewer #2: No

---

## [Editor Report · Acceptance letter]

13 Jul 2021

PONE-D-21-07914R1 

Peer video feedback builds basic life support skills: A randomized controlled non-inferiority trial 

Dear Dr. Sopka:

I'm pleased to inform you that your manuscript has been deemed suitable for publication in PLOS ONE. Congratulations! Your manuscript is now with our production department. 

Kind regards, 

on behalf of

Dr. Simone Savastano 

Academic Editor

PLOS ONE